# Checkpoint Kinase 2 Inhibition Can Reverse Tamoxifen Resistance in ER-Positive Breast Cancer

**DOI:** 10.3390/ijms232012290

**Published:** 2022-10-14

**Authors:** Ho Tsoi, Wai-Chung Tsang, Ellen P. S. Man, Man-Hong Leung, Chan-Ping You, Sum-Yin Chan, Wing-Lok Chan, Ui-Soon Khoo

**Affiliations:** 1Department of Pathology, Li Ka Shing Faculty of Medicine, The University of Hong Kong, Hong Kong SAR, China; 2Department of Clinical Oncology, Queen Mary Hospital, Hong Kong SAR, China; 3Department of Clinical Oncology, Li Ka Shing Faculty of Medicine, The University of Hong Kong, Hong Kong SAR, China

**Keywords:** breast cancer, BQ323636.1, CHK2, tamoxifen resistance, DNA damage, TMA, CCT241533, CHK2 inhibitor

## Abstract

Breast cancer is a heterogeneous disease. Tamoxifen is frequently used to treat ER-positive breast cancer. Our team has identified a novel splice variant of NCOR2, BQ323636.1 (BQ), that mediates tamoxifen resistance. However, the upstream factors that modulate BQ expression are not apparent. This study reveals that tamoxifen treatment causes induction of DNA damage which can enhance BQ expression. We show that DNA damage can activate the ATM/CHK2 and ATR/CHK1 signalling cascades and confirm that ATM/CHK2 signalling is responsible for enhancing the protein stability of BQ. siRNA or a small inhibitor targeting CHK2 resulted in the reduction in BQ expression through reduced phosphorylation and enhanced poly-ubiquitination of BQ. Inhibition of CHK2 by CCT241533 could reverse tamoxifen resistance in vitro and in vivo. Using clinical samples in the tissue microarray, we confirmed that high p-CHK2 expression was significantly associated with high nuclear BQ expression, tamoxifen resistance and poorer overall and disease-specific survival. In conclusion, tamoxifen treatment can enhance BQ expression in ER-positive breast cancer by activating the ATM/CHK2 axis. Targeting CHK2 is a promising approach to overcoming tamoxifen resistance in ER-positive breast cancer.

## 1. Introduction

Breast cancer is the leading female cancer and the second most common cause of cancer mortality in women worldwide. Breast cancer accounts for about 30% of female cancers and has a mortality-to-incidence ratio of 15% [1]. According to the World Health Organization (https://www.who.int/news-room/fact-sheets/detail/breast-cancer, accessed on 1 June 2022), in 2020, 2.3 million women had breast cancer, with 685000 deaths globally. Drug resistance, especially tamoxifen resistance, is an obstacle to breast cancer treatment. Identifying novel strategies for combating tamoxifen resistance will provide new insight into breast cancer management.

Breast cancer is heterogeneous and is characterised by the expression of oestrogen receptor α (ER), progesterone receptor (PR) and human epidermal growth factor receptor 2 (HER2). About 70% of breast cancer patients are ER-positive (ER + ve). Targeting ERs is one of the practical approaches for treating ER + ve breast cancer. There are three categories of targeted therapies available for suppressing ER signalling: selective oestrogen receptor mediator (SERM), selective oestrogen receptor degrader (SERD) and aromatase inhibitor (AI) [2]. SERMs such as tamoxifen (TAM) are antagonists that compete with oestrogen binding to ERs. TAM is the most commonly used adjuvant therapy for these patients, significantly reducing cancer recurrence and mortality [3]. However, about 50% of patients who receive TAM suffer recurrence eventually. This endocrine-resistant patient is a substantial clinical challenge.

Recurrence may be due to the development of de novo and/or acquired resistance to tamoxifen [4]. Lack of ER expression is the dominant mechanism of de novo resistance to TAM. In one of the studies, tissue microarrays were constructed from biopsy samples taken pre-treatment and at relapse from patients treated with adjuvant TAM. Out of the 29 patients who were ER + ve at pre-treatment, 5 became ER-negative at relapse [5]. Epigenetic changes in the ER gene lead to the downregulation of ERs and thus tamoxifen resistance [4]. Alterations in signalling cascades are also essential for developing acquired resistance to tamoxifen [6].

Furthermore, alternative splicing, a predominant mechanism for generating distinct mRNA isoforms from a single gene, also plays critical roles in cancer development and treatment [7]. We previously identified BQ323636.1 (BQ), a novel splice variant of NCOR2, associated with tamoxifen resistance [8]. Through various studies, BQ overexpression has been demonstrated to confer tamoxifen resistance through different mechanisms such as HIF1α signalling [9], the IL-6/STAT3 axis [9] and the IL-8/CXCR1 axis [10]. Hence, identifying the upstream modulator of BQ has become more pertinent.

It is well known that tamoxifen can induce the formation of reactive oxidative species (ROS) that will eventually induce cell death [11]. ROS can induce DNA damage [12]. TAM-resistant breast cancer cells have been reported to resist ROS elevation induced by tamoxifen [13]. We have previously shown that tamoxifen treatment could induce the expression of BQ [14], and BQ overexpression made cancer cells more resistant to ROS induction [15]. This suggests that ROS formation induced by tamoxifen modulated the expression of BQ. Thus, we hypothesise that tamoxifen can thus trigger DNA damage response (DDR) signalling by inducing ROS-mediated DNA damage. The two main signalling axes of DDR are the ataxia-telangiectasia-mutated serine/threonine kinase (ATM)/checkpoint kinase 2 (CHK2) and the ataxia-telangiectasia-mutated and Rad3-related serine/threonine kinase (ATR)/checkpoint kinase 1 (CHK1) cascades [16]. These signalling cascades might mediate the expression of BQ.

In this study, we confirmed that tamoxifen treatment would lead to the activation of CHK2, which was essential for the expression of BQ. Suppressing CHK2 could reduce nuclear BQ expression and thus reduce tamoxifen resistance in breast cancer. These findings were confirmed in clinical samples, which showed that expression of the active form of CHK2, i.e., phosphorylated CHK2 (p-CHK2), was significantly associated with nuclear BQ expression and poorer survival outcomes of ER + ve breast cancer patients. Our study, therefore, indicates the importance of DDR in modulating tamoxifen resistance and highlights that targeting DDR signalling could be a novel strategy for combating tamoxifen resistance in breast cancer.

## 2. Results

### 2.1. DNA Damage Response (DDR) Signalling Modulated the Expression of BQ in Breast Cancer

Tamoxifen (TAM) is used as adjuvant therapy for ER + ve breast cancer by repressing the activity of ERs. However, TAM has been reported to induce DNA damage [17]. Our previous studies suggest that overexpression of BQ could be an oncogenic factor in breast cancer and that TAM treatment could induce the expression of BQ [14], which made cancer cells more resistant to ROS induction [15], suggesting that DNA damage might correlate with BQ overexpression. We also found that TAM could induce ROS and DNA damage in the non-cancerous breast cell line MCF-10A, as revealed by the TUNEL assay (Appendix A). Similar to our previous publication, overexpression of BQ could compromise the effect of TAM on cell viability in MCF-10A (Appendix A). These results suggest that TAM would employ ROS and DNA damage to induce cell death.

TAM could significantly enhance cellular ROS levels in both TAM-sensitive MCF-7 and ZR-75 cell lines in a dose-dependent manner (Appendix A). Next, we employed the TUNEL assay to determine the dosage effect of TAM on DNA damage in the cell lines. As expected, TAM showed a dose-dependent effect on the degree of DNA damage; 100 nM of TAM could induce significant DNA damage in both cell lines (Figure 1A,B). To verify the results, we employed Western blot to detect p-ATM, ATM, p-ATR, ATR and ɤH2AX, which are markers of DNA damage [18,19]. The results confirmed that TAM could induce DNA damage and, thus, effectors of DDR signalling in MCF-7 and ZR-75 cells (Figure 1C and Appendix A). An amount of 100 nM of TAM could induce DNA damage signalling, as revealed by both TUNEL and Western blot assays. Next, we found that treating MCF-7 and ZR-75 cells with 100 nM of TAM could enhance BQ expression (Figure 1D and Appendix A). These results therefore are supportive that induction of DNA damage mediated by TAM could modulate the expression of BQ.

We next determined which key components in the DDR signalling cascade would be essential for BQ expression. RNAi technology was employed to reduce the expression of ATM, ATR, CHK1 and CHK2 in LCC2 and AK-47 cell lines, which are TAM-resistant cell lines with high endogenous expression of BQ. Through qPCR, we confirmed that the siRNA could significantly reduce the expression of the corresponding genes in LCC2 and AK-47 (Appendix A). We next determined the effect of these siRNAs on BQ expression. By Western blot, we confirmed that knockdown of ATM and CHK2 could significantly reduce BQ expression in LCC2 and AK-47 (Figure 2A,B, Appendix A); however, knockdown of ATR or CHK1 did not affect the expression of BQ (Figure 2C,D, Appendix A). Next, we found that the knockdown of ATM (Figure 3A,B) and CHK2 (Figure 3C,D) could reduce TAM resistance in both LCC2 and AK-47. As expected, knockdown of ATR and CHK1 did not affect TAM response (Appendix A). These results confirmed that interference with ATM and CHK2 could reduce TAM resistance by reducing BQ expression.

### 2.2. Suppression of CHK2 Activity Could Reduce Expression of BQ and Thus Tamoxifen Resistance

We next examined the effect of suppressing the activities of ATM and CHK2 on reversing TAM resistance by small molecules. First, we identified the maximum non-lethal dosage of ATM inhibitor KU-55933 (Appendix A) on non-cancerous breast cell line MCF-10A. The purpose was to determine the effect of KU-55933 on normal cells. From the cell viability assays, the maximum non-lethal dosage of these drugs was 5 nM of KU-55933. We confirmed that 5 nM of KU-55933 did not affect the cell viability of LCC2 and AK-47 (Appendix A). Subsequently, we found that adding KU-55933 would reverse tamoxifen resistance in LCC2 (Figure 4A) and AK-47 (Figure 4B). However, the results from the clonogenic assay suggested that long-term ATM inhibition would suppress cell viability in the absence of TAM (Figure 4C,D). These results suggest that the toxicity was significant; thus, KU-55933 was unsuitable for reversing TAM resistance.

Next, we determined if targeting CHK2 would be feasible for reversing TAM resistance. CCT241533 and PV109 are CHK2 inhibitors [20,21]. We examined the dosage-dependent effect on cell viability (Appendix A). An amount of 10 nM of CCT241533 and 2 µM of PV1019 were the maximum non-lethal dosages on MCF-10A. As suggested by a previous report, PV1019 at the micromolar level would have an unpredictable off-target effect [21]. Our experiments determined 2 µM as the minimal dosage for PV1019. Hence, due to the high chance of a non-specific effect, PV1019 was not used for further study. CCT241533 at 10 nM was used instead. As expected, we confirmed that treatment of CCT241533 could reverse TAM resistance in LCC2 and AK-47 cells and would not affect the cell viability during long-term treatment (Figure 5A,B). Next, we confirmed that CCT241533 could reduce BQ expression in both cell lines (Figure 5C and Appendix A). Subsequently, we established LCC2 cell lines with stable CHK2 knockdown mediated by the shRNA. Two independent stable clones were established, and it was confirmed that knockdown of CHK2 could reduce BQ expression in these clones (Figure 5D and Appendix A). By MTT assay, we confirmed that knockdown of CHK2 could make the cell lines sensitive to TAM but would abolish the effect of CCT241533 on reversing TAM resistance (Figure 5E). These results are supportive that CCT241533 targets CHK2 to modulate TAM resistance. Finally, an animal model was employed. As expected, the treatment of CCT241533 could reverse tamoxifen resistance in a dose-dependent manner (Figure 5F and Appendix A). Therefore, our study indicated that CHK2 inhibition could effectively reverse TAM resistance.

### 2.3. CHK2 Could Modulate the Protein Stability of BQ in Breast Cancer Cells

We speculated that BQ could be a substrate of CHK2. CHK2 might modulate the phosphorylation of BQ and enhance protein stability. Thus, inhibition of CHK2 would suppress the function of BQ. To investigate this, we first employed co-immunoprecipitation to determine if CHK2 could interact with BQ. The results showed that CHK2 could bind to BQ (Figure 6A and Appendix A). Next, we determined if CCT241533 would reduce the degree of phosphorylation on BQ (p-BQ). Due to the lack of an anti-p-BQ antibody, we needed to employ an indirect method to determine the level of p-BQ. A similar approach was used in our previous study [9]. We immunoprecipitated BQ and employed an anti-phos-(Ser/Thr) antibody to determine the amount of p-BQ. The results show that adding CCT241533 could significantly reduce p-BQ (Figure 6B and Appendix A). In addition, we confirmed that TAM treatment could enhance the level of p-BQ, and CHK2 inhibition could compromise such an effect (Figure 6C and Appendix A). Finally, we confirmed that TAM treatment could reduce the poly-ubiquitination of BQ while CCT241533 could enhance it (Figure 6D and Appendix A). In addition, using qPCR, we confirmed that CCT241533 did not affect mRNA expression of BQ (Appendix A), suggesting that the effect of CHK2 on BQ should be at the protein level. Our results, therefore, highlight that CHK2 can interact with BQ and phosphorylate BQ to make it more stable.

### 2.4. Clinical Significance of p-CHK2 in ER + ve Breast Cancer

We have thus shown in vitro that active CHK2 can enhance BQ expression. Our previous study confirmed that nuclear BQ expression was significantly associated with TAM resistance in breast cancer [14]. This study identified that the ATM/CHK2 axis is essential for maintaining BQ expression, with active CHK2 playing an important role in facilitating BQ overexpression. Thus, high expression of active CHK2 might be necessary for BQ overexpression and the development of TAM resistance in ER + ve breast cancer.

To demonstrate this possible correlation between CHK2 and BQ in vivo, we examined through immunohistochemistry the expression of active CHK2 (p-CHK2 Thr68; hereafter referred to as p-CHK2) in primary breast tumour tissues (Figure 7A) in the tissue microarray (TMA), correlating this with nuclear BQ expression. A Chi-square test confirmed that nuclear p-CHK2 expression was indeed significantly positively associated with nuclear BQ expression (*p* = 1.98 × 10^−9^; Figure 7B). As expected, the nuclear expression of p-CHK2 in the high nuclear BQ expression group was significantly higher than in the low nuclear BQ expression group (Mann–Whitney U test *p* = 3.35 × 10^−4^; Figure 7C). In ER + ve breast cancer cases treated with tamoxifen, correlating with clinical outcome, we found that the high nuclear p-CHK2 score was significantly associated with TAM resistance (Chi-square test *p* = 0.013; Figure 7D). Kaplan–Meier survival analyses of these cases showed that high nuclear p-CHK2 was significantly associated with poor outcomes for overall survival (Log-rank test; *p* = 2.12 × 10^−4^; Figure 7E) and disease-specific survival (Log-rank test; *p* = 0.001; Figure 7F). Univariate Cox regression analysis for overall survival (Table 1) showed cases with high nuclear p-CHK2 were significantly associated with poorer overall survival (RR = 2.555, 95% CI 1.526, 4.277; *p* = 3.59 × 10^−4^) (Table 1A), which remained significant on multivariate analysis (RR = 2.588, 95% CI 1.545, 4.333; *p* = 3.01 × 10^−4^) (Table 1B). Cox regression analysis for disease-specific survival also showed that cases with high nuclear p-CHK2 were associated with poorer disease-specific survival both for univariate with statistical significance (RR = 2.781, 95% CI 1.458, 5.305; *p* = 0.002) (Table 1A) and multivariate analyses (RR = 3.344, 95% CI 1.585, 7.085; *p* = 0.002) (Table 1B). These results suggest that nuclear p-CHK2 expression could be an independent prognostic factor in ER + ve breast cancer.

## 3. Discussion

Breast cancer is a heterogeneous disease with various molecular subtypes that require different treatments. For ER + ve breast cancer, tamoxifen (TAM) is the most commonly used drug [22]. Although TAM can effectively reduce mortality in these patients, about half the patients will develop resistance [23]. The drug resistance significantly hinders its clinical utilisation and reduction in survival for the patients. For these resistant patients, chemotherapy will be their only choice in the end. The adverse side effect of chemotherapy severely compromises the quality of life. Combating resistance may extend the usage of TAM and thus delay the administration of systematic chemotherapy.

The aberrant mechanism which promotes the development of drug resistance in many types of cancer is the PI3K/AKT pathway. Mutated PIK3CA and AKT and loss of PTEN are commonly found in breast cancer, leading to uncontrollable cell growth, proliferation, survival and non-responsiveness to TAM therapy [24,25]. The PI3K protein contains the dual activities of serine/threonine kinase and phosphatidylinositol kinase. PI3K can be activated by interacting with various growth factor receptors, such as EGFR, VEGFR and FGFR [26]. In addition, PI3K can be activated by recruiting an adaptor protein to promote the binding of p110 and p85 [27]. Activated PI3K can convert phosphatidylinositol 3,4-bisphosphate (PIP2) into 3,4,5-triphosphate (PIP3), which serves as the second messenger and binds to phosphoinositide-dependent kinase-1 (PDK1) to phosphorylate AKT [28]. AKT is the critical signal transduction protein that phosphorylates several substrates and downstream effectors. Several PI3K inhibitors, including alpelisib, idelalisib and copanlisib, have been approved by the Food and Drug Administration (FDA) [29]. PI3K/AKT inhibition combined with different endocrine drugs, such as tamoxifen and aromatase inhibitor, is a new strategy for ER + ve breast cancer treatment [30,31]. Since the PI3K/AKT pathway plays a vital role in TAM resistance [30], using these inhibitors might reduce TAM resistance. They are, however, not without challenges, such as drug-related toxicities and adverse effects [32]. Uncovering other molecular mechanisms that confer TAM resistance might provide essential information for developing new strategies to combat TAM resistance in breast cancer.

Our previous studies indicated that BQ overexpression is likely be essential for developing TAM resistance in ER + ve breast cancer, whilst downregulation of BQ by siRNA could reduce the cell viability; however, we did not determine if BQ knockdown would affect cell viability and response to TAM in TAM-sensitive breast cancer cells [14,33]. We found that overexpression of BQ could modulate the cellular activities of AR/IL-8 [10], IL-6/STAT3 [33] and HIF-1α [9] to induce TAM resistance. By interfering with these mechanisms, TAM resistance could be reduced. This also suggests that the presence of BQ would be critical, and high expression of BQ might be an indicator of a particular subtype of ER + ve breast cancer. Any agent that could reduce BQ expression should help reduce TAM resistance. Our current study confirmed that activation of DNA damage response (DDR) signalling is associated with a high expression of BQ (Figure 1). One common feature of cancer therapeutic agents is the ability to induce DNA damage, such as DNA double-strand break (DSB). Cells have the mechanism to repair DNA damage. ATM and ATR are two DNA damage sensors. Once activated, ATR and ATM signal the downstream effectors CHK1 and CHK2, respectively, to repair the damage through non-homologous end joining (NHEJ) or homologous recombination (HR) [34]. Our knockdown experiments suggest that the ATM/CHK2 axis is likely necessary for mediating BQ expression in TAM-resistant breast cancer cells, whilst that of ATR and CHK1 is not (Figure 2). These findings are the first report to demonstrate that ATM/CHK2 axis can modulate BQ expression.

Next, we confirmed that targeting ATM and CHK2 by siRNA or small inhibitors could reverse TAM resistance in the resistant cells (Figure 3, Figure 4 and Figure 5). However, long-term inhibition of ATM by KU-55933 might reduce cell viability (Figure 4C,D). We believe that inhibiting ATM would have a broad spectrum of effects in cells, inducing non-cancerous cells. This unavoidable side effect would limit the usage. Fortunately, we found that CHK2 inhibition mediated by CCT241533 matched our purpose; the chemical did affect cell viability on long-term treatment but could reverse TAM resistance (Figure 5A,B). This effect was abolished in the CHK2 knockdown cells (Figure 5E), providing evidence for the specificity of CCT241533. As expected, CCT241533 achieved a similar effect in animal studies (Figure 5F). Our results demonstrated the feasibility of using CCT241533 to reverse TAM resistance in breast cancer. Other CHK2 inhibitors might be examined to identify the best one in the future. Indeed, there is currently an ongoing Phase 1A clinical trial of CHK2 inhibitor PHI-101 (NCT04678102) which is assessing its safety and tolerability in patients with platinum resistance/refractory ovarian, fallopian tube and primary peritoneal cancer, as well as a Phase 2 single arm pilot study of the CHK1/2 inhibitor (LY2606368) (NCT02203513) in BRCA1/2 mutation-associated breast, ovarian and prostate cancers to see if it can shrink the tumours [35,36].

Our results suggest that knockdown of CHK2 or inhibition of CHK2 could reduce BQ expression. These results imply that CHK2 might be involved in regulating BQ expression. Since CHK2 is a kinase, we speculated that activation of the ATM/CHK2 axis could induce phosphorylation of BQ and thus enhance its stability. Phosphorylation is a dynamic way to regulate protein activity and a protein’s structural properties, stability and dynamics [37]. For example, it has been demonstrated in lung cancer that AKT could phosphorylate ZNF332A to stabilise this transcription factor [38]. DYRK1A could enhance the protein stability of NFATc by phosphorylation [39]. Phosphorylation of MYC at different sites could have different effects; Ser62 phosphorylation could increase its stabilisation [40]. Similar, phosphorylation of BQ affects its protein properties, such as stability. Through co-immunoprecipitation, we confirmed that CHK2 could interact with BQ (Figure 6A). Using an anti-phos-(ser/thr) antibody, we found that CCT241533 could significantly reduce the degree of phosphorylation on BQ (Figure 6B) and compromise the effect of TAM on p-BQ (Figure 6C). Through a ubiquitination assay, we confirmed that the treatment of TAM could reduce the degree of poly-ubiquitination while CCT241533 could enhance poly-ubiquitination (Figure 6D). Poly-ubiquitination is a major signal for instructing a protein to enter the degradation mechanism in cells [41]. The 26S proteasome, where proteins are degraded, can recognise the poly-ubiquitin chain. Therefore, our results suggest that phosphorylation of BQ mediated by CHK2 should regulate its stability. Although we confirmed that BQ is a novel substrate of CHK2, we could not identify which particular serine or threonine residues on BQ would be targeted by CHK2 because an antibody recognising phosphorylation of a specific site on BQ is not available on the market. Our previous study suggested that overexpression of BQ could be a robust biomarker for TAM resistance in breast cancer [14]. Studying the relationship between phosphorylation of BQ and its function in TAM resistance would be a possible direction that would lead to developing an antibody specific to particular phosphorylation. Identification of a phosphorylated BQ might further improve the performance of the biomarker.

It has long been known that threonine 68 phosphorylation mediated by ATM is required to activate the kinase activity of CHK2 during DNA damage [42]. Therefore, p-CHK2 represents active CHK2. Our results suggested that CHK2 could modulate BQ phosphorylation (Figure 6B) to make it stable to confer TAM resistance. Therefore, we speculated that p-CHK2 would be associated with BQ and, thus, TAM resistance. Through in vivo study, we confirmed that the expression of p-CHK2 was significantly positively correlated with BQ expression (Figure 7B). This result was as expected, in line with in vitro experiments that indicate CHK2 can modulate the expression of BQ. Clinical samples further validated this finding, highlighting the clinical significance of p-CHK2. Since p-CHK2 would be an upstream mediator of BQ, a high nuclear p-CHK2 expression would correlate with a high nuclear BQ expression. We also observed p-CHK2 was associated with TAM resistance (Figure 7D). Overall and disease-specific survival analyses confirmed that the patients with high nuclear p-CHK2 expression had poorer survival outcomes (Figure 7E,F). Cox regression analysis further confirmed this finding by both univariate and multivariate analysis (Table 2). These results suggest that p-CHK2 could be another independent prognostic factor, illustrating the clinical significance of p-CHK2 in ER + ve breast cancer. In Table 2, high nuclear BQ expression is also significantly associated with poorer overall and disease-specific survival, re-confirming our previous reports [8,14]. We also observed that the patients with high expression of both BQ and p-CHK2 have in fact the highest risk ratio in both overall and disease-specific survival analyses for both univariate analysis (RR = 5.196 CI 95% 2.395, 11.271; *p* = 3.04 × 10^−5^) (RR = 4.944 CI 95% 2.016, 12.127; *p* = 4.81 × 10^−4^) as well as for multivariate analysis (RR = 5.181 CI 95% 2.388, 11.242; *p* = 3.14 × 10^−5^) (RR = 5.393 CI 95% 1.961, 14.831; *p* = 0.001). This indicates that p-CHK2 and BQ have an additive effect in enhancing breast cancer development. p-CHK2 is likely to employ pathways other than BQ to modulate disease, while BQ has other upstream factors to enhance its expression. By identifying these factors, we will better understand TAM resistance in ER + ve breast cancer.

In summary, we confirmed that TAM treatment could enhance BQ expression by activating the ATM/CHK2 axis. Activated CHK2 could stabilise BQ, which could confer TAM resistance. Targeting CHK2 could reverse TAM resistance in vitro and in vivo. In vivo study confirmed that high nuclear expression of p-CHK2 was positively correlated with nuclear BQ expression and was significantly associated with TAM resistance and poorer survival outcome. Therefore, our current study confirmed the clinical significance of p-CHK2 and illustrated that targeting CHK2 could be a possible strategy for combating TAM resistance in breast cancer.

## 4. Materials and Methods

### 4.1. Cell Culture, Transfection and Stable Cell Line Establishment

Human non-tumorigenic breast cell line MCF-10A and breast cell lines MCF-7 and ZR-75 (tamoxifen-sensitive cell lines) were obtained from American Type Culture Collection (ATCC, Manassas, VA, USA). They were re-authenticated by short tandem repeat profiling [14]. Tamoxifen-resistant cell line LCC2 is derived from MCF-7. AK47 is a tamoxifen-resistant cell line derived from ZR-75. LCC2 and AK-47 were kindly provided by Dr. Robert Clarke (Georgetown University Medical School, Washington, DC, USA) and used in our previous study [14]. MCF-7-BQ and ZR-75-BQ, stably transfected by mammalian expression plasmid pcDNA3.1-His-BQ, were used [14]. MCF-10A was cultured in DMEM/F12 (11330032; ThermoFisher, Waltham, MA, USA) containing 5% horse serum (16050122; Thermo Fisher Scientific, Waltham, MA, USA), 100 ng/mL Cholera Toxin (C-8052; Sigma-Aldrich, St. Louis, MO, USA), 0.5 mg/mL Hydrocortisone (H-0888; Sigma-Aldrich, St. Louis, MO, USA), 10 μg/mL insulin (I-1882; Sigma-Aldrich, St. Louis, MO, USA), 20 ng/mL EGF (PHG0313; Thermo Fisher Scientific, Waltham, MA, USA) and 1% antibiotics penicillin/streptomycin (P/S; 10378016; Thermo Fisher Scientific, Waltham, MA, USA). MCF-7, MCF-7-BQ and LCC2 cells were cultured and maintained in DMEM (12100046; Thermo Fisher Scientific, Waltham, MA, USA) containing 10% FBS (26140079; Thermo Fisher Scientific, Waltham, MA, USA) and 1% P/S (10378016; Thermo Fisher Scientific, Waltham, MA, USA). ZR-75, ZR-75-BQ and AK-47 cells were cultured in IMEM (A104890; Thermo Fisher Scientific, Waltham, MA, USA) with 10% FBS and 1% P/S. The cell lines were cultured in the incubator at 37 °C supplied with 5% CO_2_. Lipofectamine 2000 reagent (11668019; Thermo Fisher Scientific, Waltham, MA, USA) was used for the transfection of plasmids according to the manufacturer’s instructions. After 72 h post-transfection, 0.5 µg/mL puromycin (A1113802; Thermo Fisher Scientific, Waltham, MA, USA) was used to select the transfected cells. Fresh DMEM or IMEM containing 10% FBS, 1% P/S and 0.5 µg/mL of puromycin was replaced every 72 h. The selection was performed for 6 weeks. The cells were incubated in DMEM or IMEM containing 10% FBS, 1% P/S and 0.5 µg/mL of puromycin. Oligofectamine reagent (12252011; Thermo Fisher Scientific, Waltham, MA, USA) was used for siRNA transfection according to the manufacturer’s instructions. Cell lines used were confirmed mycoplasma-free. The mycoplasma screening was performed by the Faculty Core Facility (LKS Faculty of Medicine, The University of Hong Kong, Hong Kong SAR) to ensure the cell culture was free from mycoplasma.

### 4.2. Plasmids, siRNA, shRNA and RT-qPCR

pcDNA3.1-His-BQ323636.1 was used. CHK2 human shRNA plasmid (sc-29271-SH; Santa Cruz Biotechnology, Dallas, TX, USA) was employed, and control shRNA (sc-108080; Santa Cruz Biotechnology, Dallas, TX, USA) was purchased. siRNA against ATM (L-003201), ATR (L-003202), CHK1 (L-003255) and non-targeting siRNA (D-001810) were purchased from Horizon Discovery (Cambridge, UK). Total RNA was isolated using TRIzol (15596026; ThermoFisher, Waltham, MA, USA) according to the manufacturer’s instructions. cDNA synthesis was generated using PrimeScript^TM^ RT Master Mix (RR036B; Takara Biomedical Technology Co., Ltd., China). RNA in the amount of 0.5 µg was used for cRNA synthesis. qPCR was performed using the StepOne Real-Time PCR system (Thermo Fisher Scientific, Waltham, MA, USA). qPCR was using PowerUp™ SYBR™ Green Master Mix (A25742; ThermoFisher, Waltham, MA, USA) and qPCR primers. The following primers (5′→3′) were used: ATM-F (CTG CAC ACA AGC CCA TTC TT), ATM-R (AGG AAG TGT GTT TGC CT), ATR-F (TGA TGG GTC ATG CTG TGG AA), ATR-R (ACT CAT CAA CTG CAA AGG AGC), BQ-F (GGA GCG CAT GCA GAG AAC C), BQ-R (CTG GCG GTC TTT GTA CAC CT), CHK1-F (TCA TGG CAG GGG TGG TTT AT), CHK1-R (GTT GCC AAG CCA AAG TCT GA), CHK2-F (AAA CTC CAG CCA GTC CTC TC), CHK2-R (AAA CTC CAG CCA GTC CTC TC), GAPDH-F (GCA AAT TCC ATG GCA CCG T) and GAPDH-R (TCG CCC CAC TTG ATT TTG G). The relative gene expression was determined using ΔΔCT method with GAPDH as the internal control.

### 4.3. Western Blot

Cells were lysed in cell lysis buffer (9803; Cell Signaling Technology, Danvers, MA, USA) supplemented with the protease inhibitor cocktail (4693159001; Sigma-Aldrich, St. Louis, MO, USA). Protein concentrations were tested by DC protein assay kit (5000112; Bio-Rad, Hercules, CA, USA). An amount of 20 μg of protein sample was used. Proteins were separated by SDS-PAGE and transferred to PVDF (1620177; Bio-Rad, Hercules, CA, USA). The following antibodies were used: anti-p-ATM (Ser1981) (1:1000; 5883; Cell Signaling Technology, Danvers, MA, USA), anti-ATM (1:1000; 2873; Cell Signaling Technology, Danvers, MA, USA), anti-p-ATR (Ser1989) (1:1000; 30632; Cell Signaling Technology, Danvers, MA, USA), anti-ATR (1:1000; 2790; Cell Signaling Technology, Danvers, MA, USA), anti-BQ (1:1000; D-12, Veritech Ltd., Hong Kong), anti-p-CHK1 (Ser345) (1:1000; 2348; Cell Signaling Technology, Danvers, MA, USA), anti-CHK1 (1:1000; 2360; Cell Signaling Technology, Danvers, MA, USA), anti-p-CHK2 (Thr68) (1:1000; 2197; Cell Signaling Technology, Danvers, MA, USA), anti-CHK2 (1:1000; 3440; Cell Signaling Technology, Danvers, MA, USA), anti-HIS tag (1:5000; 018-23224; FUJIFILM Wako Pure Chemical Corporation, Osaka, Japan), anti-ɤH2AX (Ser139) (1:2000; 9718; Cell Signaling Technology, Danvers, MA, USA), anti-ubiquitin (1:500; sc-166553; Santa Cruz Biotechnology, Dallas, TX, USA) and anti-GAPDH (1:5000; sc-365062; Santa Cruz Biotechnology, Dallas, TX, USA). Anti-mouse HRP (1:5000; P0447; Agilent Dako, Santa Clara, CA, USA) and anti-rabbit HRP (1:5000; P0260; Agilent Dako, Santa Clara, CA, USA) and Protein A-HRP (18-160; Sigma-Aldrich, St. Louis, MO, USA) were used as secondary antibodies.

### 4.4. Co-Immunoprecipitation

An amount of 1 × 10^6^ of LCC2 or AK-47 cells were transfected with 2 ug of pcDNA3.1-His-BQ323636.1 using lipofectamine2000 (11668019; Thermo Fisher Scientific, Waltham, MA, USA). Cells were harvested after 48 h of post-transfection. Cells were lysed in 200 µL of ice-cold Co-IP buffer (20 mM Tris-Cl, pH 7.4, 100 mM NaCl, 5 mM MgCl_2_, 0.5% NP-40, 10% glycerol); 10 µL of the cell lysate was used as input, 95 µL of the lysate was subjected to immunoprecipitation with anti-His (1: 200; MA1-21315; Thermo Fisher Scientific, Waltham, MA, USA), anti-CHK2 (1:200; 6334; Cell Signaling Technology, Danvers, MA, USA), anti-mouse-IgG (1:200; X0931; Agilent Dako, Santa Clara, CA, USA) or anti-rabbit IgG (1:200; X0903; Agilent Dako, Santa Clara, CA, USA) and 50 µL of Dynabeads™ Protein A (10002D; Thermo Fisher Scientific, Waltham, MA, USA) was used. The antibody was incubated with the beads at 4 °C for 2 h with rotation and then subsequently washed three times with 1 mL of Co-IP buffer. The beads were then incubated with the cell lysates at 4 °C overnight with rotation. The loaded beads were washed with 1 mL of Co-IP buffer for 10 min three times at room temperature. The proteins were eluted by incubating the beads with 50 µL of 2× SDS sample buffer at 99 °C for 5 min. The following antibodies were employed: anti-his (1:5000; PA1-983B; Thermo Fisher Scientific, Waltham, MA, USA), anti-BQ (1:1000; D-12, Veritech Ltd., Hong Kong), anti-CHK2 (1:1000; 3440; Cell Signaling Technology, Danvers, MA, USA), anti-phos-(Ser/Thr) Antibody (1:2000; 9631; Cell Signaling Technology, Danvers, MA, USA).

### 4.5. Cell Viability and Functional Assays

MTT assay (M6494; ThermoFisher, Waltham, MA, USA) was performed. First, 5000 cells were seeded in 96-well plates. A clonogenic assay was performed with 2000 cells seeded in 12-well plates. The cells were stained with 0.01% crystal violet (C0775; Sigma-Aldrich, St. Louis, MO, USA). ROS level was determined by CM-H_2_DCFDA (C6827; ThermoFisher, Waltham, MA, USA). TUNEL assay kit (ab66108; Abcam, Cambridge, UK) was employed to determine DNA damage. Absorbance and fluorescent signals were recorded by microplate reader Infinite F200 (Tecan, Seestrasse, Switzerland). All experiments were performed in triplicate.

### 4.6. Chemicals

CCT241533 (CHK2 inhibitor; HY-14715; MedChemExpress LLC, Monmouth Junction, NJ, USA), PV1019 (CHK2 inhibitor; 220488; Sigma, St. Louis, MO, USA), KU-55933 (ATM inhibitor; S1092; Selleck Chemicals LLC, Houston, TX, USA), MG-132 (S2619; Selleck Chemicals LLC, Houston, TX, USA), 4-hydroxytamoxifen (TAM; H7904; Sigma, St. Louis, MO, USA) were purchased. These chemicals were dissolved into DMSO (D8418; Sigma, St. Louis, MO, USA).

### 4.7. Xenograft

Five- to six-week-old female nude mice (strain: BALB/cAnN-nu;) were used for this study. The cell mixture was made by mixing 50 µL of the cell suspension containing 1 × 10^6^ LCC2 cells with 50 µL of Matrigel (356234; BD Bioscience, Franklin Lakes, NJ, USA). Then, 100 µL of the mixture was injected into the mice’s abdominal mammary fat pad. Mice were randomised into 5 groups when the tumours were palpable: (1) saline (N = 5); (2) TAM (N = 5); (3) 1 mg/Kg CCT241533 (N = 5); (4) TAM + 2.5 mg/Kg CCT241533 (N = 5); (5) TAM + 5 mg/Kg CCT241533 (N = 5). Saline or TAM consisting of 0.5 mg of tamoxifen dissolved in peanut oil (C2144; Sigma; St. Louis, MO, USA), CCT241533 or TAM + CCT241533 were given by subcutaneous injection 2 times (Monday and Thursday) per week for 6 weeks. In total, the mice received 12 injections. The tumour sizes were measured regularly using an electronic calliper, and the tumour volume was determined using the formula: the longest diameter × (shortest diameter)^2^/2. All the procedures were approved by the HKU Committee on the Use of Live Animals in Teaching and Research (5230-19).

### 4.8. Tissue Microarray and Immunohistochemistry (IHC)

Approval (HKU/HA HKW IRB No. UW 08-147) for clinical investigation was obtained from the Institutional Review Board of The University of Hong Kong and Hospital Authority Hong Kong West Cluster. Clinical information was obtained from the Department of Pathology, Queen Mary Hospital of Hong Kong. Histological sections were reviewed by the pathologists. For individual tumours, donor blocks were chosen from the representative paraffin tumour blocks. The selected region was selected for the constructing tissue microarray (TMA). Tamoxifen resistance was defined as patients treated with tamoxifen in the adjuvant setting but subsequently developed disease relapse or distant metastases. Only cases with a clear history of treatment response were used to analyse tamoxifen resistance. A total of 313 breast cancer cases (Table 2) were included. There were 185 ER + ve cases, which were used for scoring BQ323636.1 and p-CHK2 staining. Each case was constructed as three replicates, and the average score was used for the case. TMA sections were deparaffinised by xylene and rehydrated by ethanol. Antigen retrieval process was performed with 0.01 M citrate buffer (pH 6.0). The slides were immersed in the solution containing 3% H_2_O_2_ to quench endogenous peroxidase. The slides were rinsed with TBST twice, followed by incubation with primary monoclonal anti-p-CHK2 (Thr68) antibody (p-CHK2; 1:50; 2197; Cell Signaling Technology, Danvers, MA, USA) and BQ323636.1 antibody (1:50; D-12, Veritech Ltd., Hong Kong) at 4 °C overnight. The slides were washed further by TBST and incubated with Envision + System-HRP Labelled Polymer Anti-Rabbit (K4003; Agilent Dako, Santa Clara, CA, USA) and Anti-Mouse (K4001; Agilent Dako, Santa Clara, CA, USA) for 30 min at room temperature. The slides were then washed by TBST, followed by incubation with chromogen DAB/substrate reagent for 1 min. After dehydration, the slides were mounted. TMA slides were visualised by the Aperio ScanScope system (Leica Biosystems, Wetzlar, Germany). Scoring was performed by two individuals. H-score was used for p-CHK2 and BQ323636.1 expression. It was calculated as follows: 1 × % of cells stained at low intensity + 2 × % of cells stained at moderate intensity + 3 × % of cells stained at high intensity. The median of the H-score was used as the threshold, which was 110 for nuclear BQ and 2 for nuclear p-CHK2, respectively.

### 4.9. Statistical Analyses

All data were processed in Excel 2016 (Microsoft), Prism5 (GraphPad) or SPSS25 (IBM). Mean value ± SD from at least three independent experiments was used to express the data. Student’s *t*-tests were performed to determine the significance between the 2 groups. The statistical significance between two groups from clinical samples was determined by Mann–Whitney U test. The tests were two-sided. One-way ANOVA and two-way ANOVA with Bonferroni’s post-test were employed for the comparisons with multiple groups. Chi-square (χ2) test was employed to reject the null hypothesis. Survival analysis was performed using Kaplan–Meier estimates followed by a Log-rank test. Cox regression was used to estimate the association between clinical–pathological parameters, nuclear BQ score, nuclear p-CHK2 score and survival. Relative risk (RR) with 95% confidence interval (CI) were determined. The proportional-hazards assumption was tested using the Omnibus test, and no major model violation was observed: *, ** and *** in the figures indicated *p* < 0.05, *p* < 0.01 and *p* < 0.001, respectively.

## 5. Conclusions

Our results demonstrate that high expression of CHK2 was associated with high nuclear expression of BQ in ER + ve breast cancer. Suppressing the activity of CHK2 could reduce the protein stability and expression of BQ, resulting in the reduction in TAM resistance. Our findings suggest that targeting CHK2 could be a novel strategy for combating TAM resistance in ER + ve breast cancer.

## Figures and Tables

**Figure 1 ijms-23-12290-f001:**
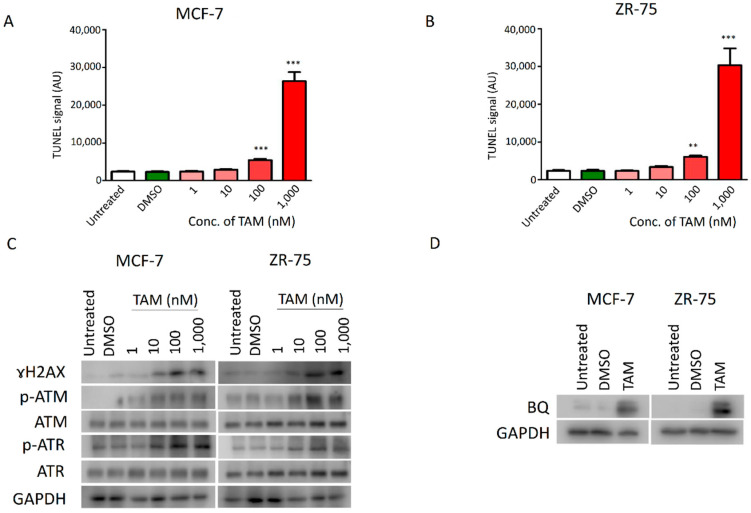
TAM could induce DNA damage and BQ expression. Dose-dependent effect of TAM on the induction of DNA damage in (**A**) MCF-7 and (**B**) ZR-75. The cells were incubated with different dosages of TAM for 72 h. TUNEL assay was employed to determine DNA damage. The fluorescence signal was recorded. (**C**) The effect of TAM on the expression of ɤH2AX (Ser139), p-ATM (Ser1981), ATM, p-ATR (Ser1989) and ATR were determined by Western blot. The cells were treated for 72 h. The expression of candidate proteins was detected by Western blot. GAPDH was used as the loading control. (**D**) The effect of TAM on BQ expression. The cells were incubated with 100 nM of TAM for 72 h. Western blot was used for the detection of endogenous BQ. GAPDH was used as the loading control. Results are shown as mean ± SD from four independent experiments. One-way ANOVA with Bonferroni’s post-test was employed to compare the statistical significance with the untreated control: ** and *** represents *p* < 0.01 and *p* < 0.001, respectively.

**Figure 2 ijms-23-12290-f002:**
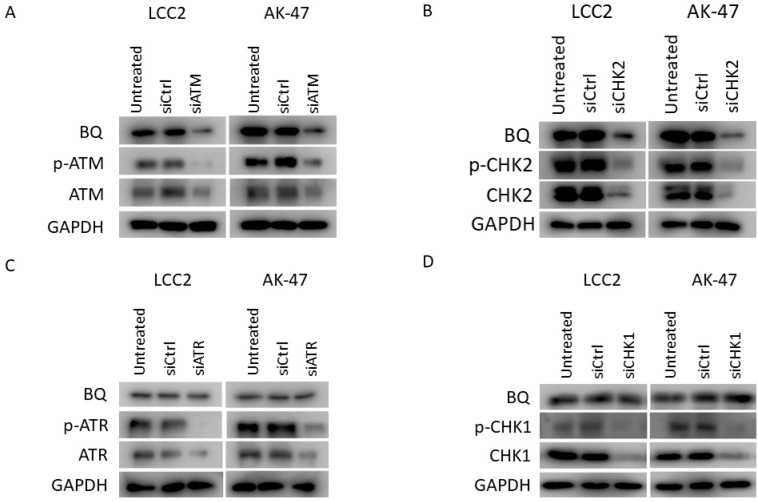
Knockdown of ATM and CHK2 could reduce BQ expression. (**A**) The effect of ATM knockdown on pATM (Ser1981) and BQ expression. (**B**) The effect of CHK2 knockdown on p-CHK2 (Thr68) BQ expression. (**C**) The effect of ATR knockdown on p-ATR (Ser1989) BQ expression. (**D**) The effect of CHK1 knockdown on p-CHK1 (Ser345) and BQ expression. LCC2 and AK-47 cells were transfected with 20 nM of the siRNA. Cells were harvested 72 h post-transfection. Western blot was employed to detect the expression of candidate proteins. GAPDH was used as the loading control.

**Figure 3 ijms-23-12290-f003:**
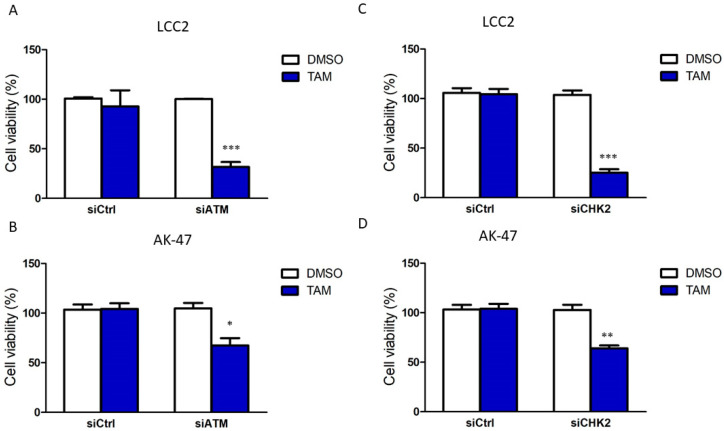
Knockdown of ATM and CHK2 could reduce TAM resistance. The effect of ATM knockdown on TAM response in (**A**) LCC2 and (**B**) AK-47. The effect of CHK2 knockdown on TAM response in (**C**) LCC2 and (**D**) AK-47. LCC2 and AK-47 are TAM-resistant cell lines. The cells were incubated with 20 nM of the siRNA and 4 µM of TAM for 96 h. Cell viability was determined by MTT assay. Results are shown as mean ± SD from six independent experiments. Student’s *t*-test was employed to compare the statistical significance between DMSO and TAM groups: * *p* < 0.05, ** *p* < 0.01 and *** *p* < 0.001.

**Figure 4 ijms-23-12290-f004:**
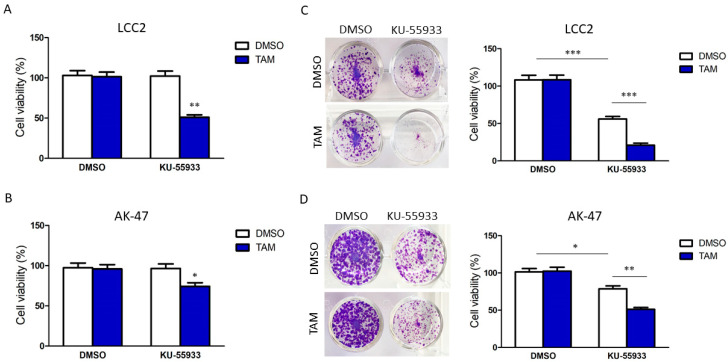
The effect of ATM inhibition by KU-55933 on TAM resistance and cell viability. The short-term effect of KU-55933 on TAM response in (**A**) LCC2 and (**B**) AK-47. The cells were treated with 5 nM of KU-55933 and 4 µM of TAM for 96 h. Cell viability was determined by MTT assay. Student’s *t*-test was employed to compare the statistical significance between DMSO and TAM groups. The long-term effect of KU-55933 on TAM response in (**C**) LCC2 and (**D**) AK-47. The cells were treated with 5 nM of KU-55933 and 4 µM of TAM for 14 days. A clonogenic assay was employed to determine cell viability. Results are shown as mean ± SD from six independent experiments. Student’s *t*-test was employed to compare the statistical significance between indicated groups: * *p* < 0.05, ** *p* < 0.01 and *** *p* < 0.001.

**Figure 5 ijms-23-12290-f005:**
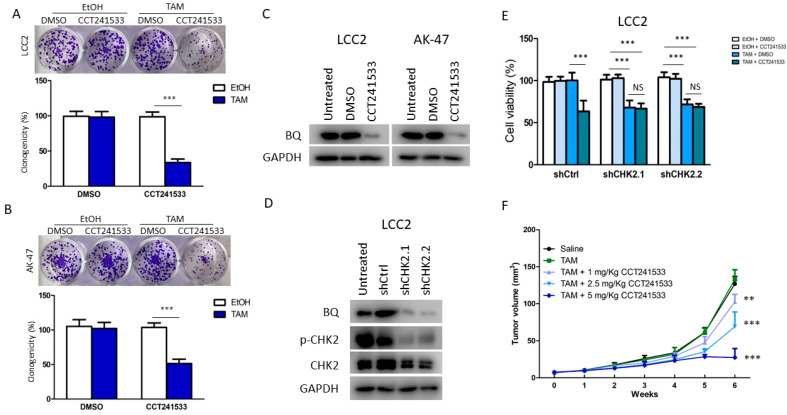
Targeting CHK2 by CCT241533 could reverse TAM resistance. The long-term effect of KU-55933 on TAM response in (**A**) LCC2 and (**B**) AK-47. The cells were treated with 10 nM of CCT241533 and 4 µM of TAM for 14 days. A clonogenic assay was employed to determine cell viability. Student’s *t*-test was employed to compare the statistical significance between DMSO and TAM groups. (**C**) The effect of CCT241533 on BQ expression. LCC2 and AK-47 were treated with 10 nM of CCT241533 for 72 h. Western blot was employed to determine BQ expression. GAPDH was used as the loading control. (**D**) The knockdown efficiency of shRNA against CHK2. LCC2 cells were stably transfected to establish cell lines with stable CHK2 knockdown. Two independent stable clones were selected. Western blot was employed to detect the expression of BQ, p-CHK2 (Thr68) and CHK2. GAPDH was used as the loading control. (**E**) Knockdown of CHK2 could abolish the effect of CCT241533 on reversing TAM resistance. LCC2 and AK-47 were treated with 10 nM of CCT241533 and 4 µM of TAM for 96 h. Cell viability was determined by MTT assay. One-way ANOVA with Bonferroni’s post-test was employed to compare the statistical significance with indicated groups. (**F**) CCT241533 could reduce TAM resistance in female nude mice. LCC2 cells were used for establishing xenografts. The mice were treated with saline, TAM (0.5 mg/Kg) and CCT241533 (1 mg/Kg, 2.5 mg/Kg and 5 mg/Kg) via subcutaneous injection. The mice were treated twice per week for six weeks. The tumour growth curve was plotted. Two-way ANOVA with Turkey post-test was employed to compare the statistical significance with the saline group: ** *p* < 0.01 and *** *p* < 0.001.

**Figure 6 ijms-23-12290-f006:**
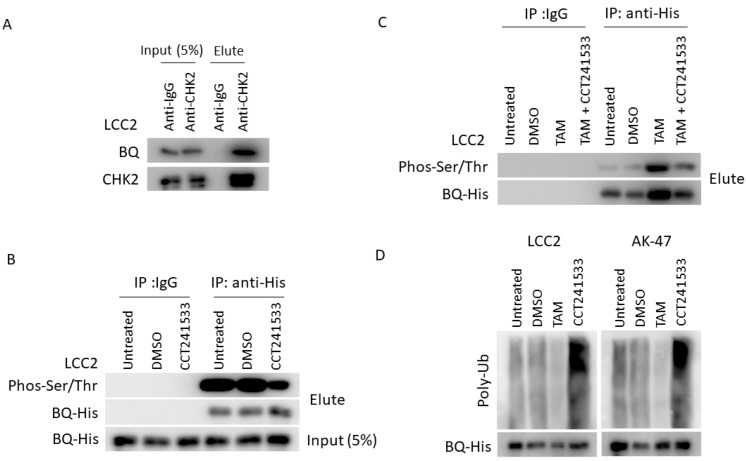
CHK2 could phosphorylate BQ to enhance its protein stability. (**A**) CHK2 could interact with BQ. LCC2 cells were lysed and immunoprecipitated with anti-CHK2. The immunoprecipitant was analysed by Western blot to detect the presence of endogenous BQ and CHK2. (**B**) The treatment of CCT241533 could reduce the degree of BQ phosphorylation. LCC2 cells were transfected with pcDNA3.1-His-BQ, and the cells were treated with 10 nM of CCT241533 for 48 h. The cells were immunoprecipitated with anti-His antibody. The immunoprecipitant was analysed with anti-His to confirm the presence of BQ in the immunoprecipitant and anti-phos-(Ser/Thr) antibody to determine the degree of phosphorylation on BQ protein. (**C**) The addition of CCT241533 could reduce the relative level of p-BQ in TAM-treated LCC2 cells. LCC2 cells were transfected with pcDNA3.1-His-BQ, and the cells were treated with 4 µM of TAM and 10 nM of CCT241533 for 48 h. The cells were immunoprecipitated with anti-His antibody. The immunoprecipitant was analysed with protein A-HRP and anti-phos-(Ser/Thr) antibody to determine the degree of phosphorylation on BQ protein. (**D**) The effect of TAM and CCT241533 on poly-ubiquitination of BQ. LCC2 cells were transfected with pcDNA3.1-His-BQ, and the cells were treated with 5 µM of MG132 together with 4 µM of TAM or 10 nM of CCT241533 for 24 h. The cells were immunoprecipitated with anti-His antibody. The immunoprecipitant was analysed with anti-His to confirm the presence of BQ in the immunoprecipitant and anti-Ub antibody to determine the degree of poly-ubiquitination on BQ protein.

**Figure 7 ijms-23-12290-f007:**
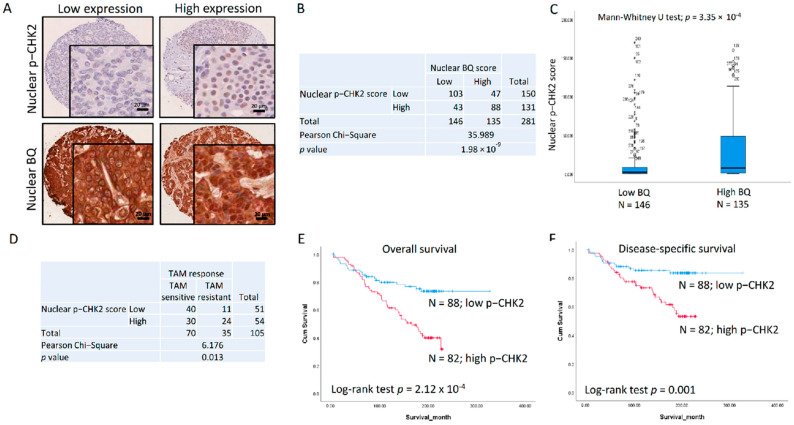
The clinical significance of p-CHK2 in breast cancer. (**A**) The expression of p-CHK2 and BQ in primary breast tumours on tissue microarray (TMA). Immunohistochemistry was performed on TMA to determine the expression of p-CHK2 and BQ in the tissues. The expression of nuclear p-CHK2 and nuclear BQ were scored. (**B**) Nuclear p-CHK2 was positively correlated with nuclear BQ. Chi-square test was performed. (**C**) Nuclear expression of p-CHK2 in the tumours with the high nuclear BQ expression group was significantly higher than that in the low nuclear BQ expression group. Mann–Whitney U test was employed to determine statistical significance. (**D**) Nuclear p-CHK2 was positively correlated with TAM resistance in ER + ve cases. Chi-square test was performed. ER + ve patients with high nuclear expression of p-CHK2 had poorer outcomes for (**E**) overall and (**F**) disease-specific survival. Log-rank test was employed to determine statistical significance.

**Table 1 ijms-23-12290-t001:** (**A**) Univariate Cox regression analysis of overall and disease-specific survival in ER + ve breast cancer. (**B**) Multivariate Cox regression analysis of overall and disease-specific survival in ER + ve breast cancer.

**(A)**					
		**Overall Survival**		**Disease-Specific Survival**	
**Clinical–Pathological Parameters**	**Cases**	**RR (95% CI)**	***p* Value**	**RR (95% CI)**	***p* Value**
Age	183	1.994 (1.241, 3.204)	0.004	1.102 (0.618, 1.967)	0.742
T stage	133	1.579 (0.564, 4.423)	0.384	1.150 (0.273, 4.852)	0.849
Lymph node involvement	169	1.446 (0.870, 2.404)	0.155	2.148 (1.116, 4.137)	0.022
Tumour grade	174	1.357 (0.834, 2.208)	0.220	2.200 (1.198, 4.041)	0.011
Histological type	183	1.020 (0.536, 1.942)	0.952	0.699 (0.347, 1.406)	0.315
PR status	169	0.686 (0.384, 1.225)	0.202	0.484 (0.251, 0.931)	0.030
HER2 status	140	1.084 (0.527, 2.226)	0.827	1.114 (0.459, 2.702)	0.812
Tumour size	65	1.417 (0.581, 3.455)	0.444	2.248 (0.623, 8.110)	0.216
High nuclear pCHK2 score	170	2.555 (1.526, 4.277)	3.59 × 10^−4^	2.781 (1.458, 5.305)	0.002
High nuclear BQ score	171	2.713 (1.620, 4.545)	1.49 × 10^−4^	2.897 (1.519, 5.527)	0.001
Both high nuclear pCHK2 and BQ score	103	5.196 (2.395, 11.271)	3.04 × 10^−5^	4.944 (2.016, 12.127)	4.81 × 10^−4^
**(B)**					
		**Overall Survival**		**Disease-Specific Survival**	
**Clinical–Pathological Parameters**	**Cases**	**RR (95% CI)**	***p* Value**	**RR (95% CI)**	***p* Value**
Age	170	1.919 (1.187, 3.103)	0.008		
High nuclear pCHK2 score	170	2.588 (1.545, 4.333)	3.01 × 10^−4^		
Age	171	1.953 (1.208, 3.156)	0.006		
High nuclear BQ score	171	2.721 (1.624, 4.559)	1.43 × 10^−4^		
Age	103	1.361 (0.736, 2.518)	0.326		
Both high nuclear pCHK2 & BQ score	103	5.181 (2.388, 11.242)	3.14 × 10^−5^		
Lymph node involvement	142			2.509 (1.226, 5.134)	0.012
Tumour grade	142			1.701 (0.845, 3.426)	0.137
PR status	142			0.391 (0.190, 0.805)	0.011
High nuclear pCHK2 score	142			3.344 (1.585, 7.058)	0.002
Lymph-node involvement	141			2.369 (1.133, 4.952)	0.022
Tumour grade	141			1.795 (0.908, 3.549)	0.093
PR status	141			0.384 (0.183, 0.807)	0.012
High nuclear BQ score	141			2.639 (1.286, 5.413)	0.008
Lymph node involvement	89			2.711 (1.112, 6.610)	0.028
Tumour grade	89			1.277 (0.533, 3.060)	0.583
PR status	89			0.441 (0.172, 1.130)	0.088
Both high nuclear pCHK2 and BQ score	89			5.393 (1.961, 14.831)	0.001

**Table 2 ijms-23-12290-t002:** Clinical characterisation of breast cancer patients.

Clinical Characters		Number of CASES	Percentage (%)
Breast cancer patients		313	100
Age	<54	161	51.4
	≥54	150	47.9
T stage	I, II	180	57.5
	III, IV	17	5.4
Lymph Node status	Positive	147	47.0
	Negative	132	42.2
Tumour Grade	1, 2	131	41.9
	3	148	47.3
Tumour Size	<2 cm	39	12.5
	≥2 cm	71	22.7
Oestrogen Receptor status	Positive	185	59.1
	Negative	66	21.1
Progesterone receptor status	Positive	142	45.4
	Negative	95	30.4
HER2 receptor status	Positive	38	12.1
	Negative	158	50.5
Triple Negative status	Positive	36	11.5
	Negative	189	60.4
pCHK2	No expression	77	24.6
	Expression	236	70.0

## Data Availability

The materials and resources in this study are available from the corresponding author upon reasonable request.

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
