# Peer review of "Checkpoint Kinase 2 Inhibition Can Reverse Tamoxifen Resistance in ER-Positive Breast Cancer"

_ijms, 2022, doi:10.3390/ijms232012290_

Round 1

Reviewer 1 Report

Tsoi et al have proposed a mechanism of action demonstrating the importance of CHK2 inhibition which in turn inhibits/destabilizes the function of a splice variant of NCOR2, named BQ which induces resistance to tamoxifen (TAM) in ER+ve breast cancer (BC). TAM administration causes free reactive oxygen species (ROS) that causes DNA damage but activation of CHK2 leads to overexpression of BQ which suppresses ROS function. By challenging with CHK2 inhibitor CCT241533, the team has mechanistically showcased reduction of tumour growth in vitro and in vivo and hence claimed that inhibition of CHK2 is critical in overcoming TAM resistance. Similarly, using clinical samples they have shown that samples with high phosphor-CHK2 (active form) harbour more BQ expression in the nucleus are associated with poor survival and demonstrated resistance to TAM. The major concern as per this reviewer is the size/representation of the images being too small to interpret or visualize for a readers. Though the claims are well made, addressing the following issues will benefit the message given by the authors:

11.  Does BQ overexpression (OE) is induced only TAM induction as discussed in Fig1 or is it subjected to overexpress because of DNA damage due to upregulation of p-CHK2. Does radiation also induce BQ OE?

22. Does OE of BQ in MCF10A makes it resistant to TAM?

33. The figures are too small and should be properly arranged for easier visualization

44. The authors should also show expression of total ATM, ATR along with p-CHK1, p-CHK2, total CHK2 and CHK1.

55. The authors should test TAM induction in near normal BC cells such as MCF10A to demonstrate the specific association of TAM induced upregulation in ER positive cells.

66. The authors should demonstrate the long-term effects using 3D acini culture system.

77. Does inhibition of BQ alone reduce cell viability in TAM resistant or non-resistant ER+ve BC cells? If so, a statement should me made.

88. Fig 5D: The authors claim that CHK2 shRNA KD stable cells were generated, but the results show that only BQ expression was reduced but not CHK2? Please explain?

99. Fig 5D: Rather saying two independent cell lines were established, its better to state that two independent clones of the cells line were established using respective shRNA constructs that demonstrated the KD of the gene of interest.

110. Please describe the nature of the animal model that was utilised with the proper description about the timing of treatment and monitoring conditions.

111. Fig6: Can authors prove that p-BQ is upregulated in presence of TAM and is reduced in combination of TAM and CHK2 inhibitor?

Reviewer 2 Report

The manuscript by Tsoi et al. displays a good experimental design and provided evidence is well-founded. The subject, also, could be of interest to scientists working in several fields.

The Authors used only two cell lines in this study, and this parly weakens the strenght of the work since cell lines grown in serum raise many doubts as a reliable model.

However, the points to be addressed before publication in IJMS are:

Major:

English language check

An in vitro test confirming ROS formation following TAM treatments should be shown.

Minor:

line 210: CHK2 inhibition would reduce BQ expression or activity? The sentence should be checked.

Author Response

The manuscript by Tsoi et al. displays a good experimental design and provided evidence is well-founded. The subject, also, could be of interest to scientists working in several fields.

Response:

Thank you for your appreciation.

The Authors used only two cell lines in this study, and this parly weakens the strenght of the work since cell lines grown in serum raise many doubts as a reliable model.

Response:

Thank you very much for your suggestions. We will bear this issue in mind. Instead of using nude mice (xenograft), we will consider using  patient-derived xenograft models instead. We are also generating more drug-resistant cell lines, particularlymore TAM-resistant cell lines for use in the future.

However, the points to be addressed before publication in IJMS are:

Major:

Comment 1

English language check

Response:

Thank you for your comment. We have checked the grammar and sentences to improve the language.

Comment 2

An in vitro test confirming ROS formation following TAM treatments should be shown.

Response:

Thank you for your comment. We have employed CM-H2DCFDA to determine the level of ROS induced by tamoxifen in MCF-7 and ZR-75 cells. We have included this result in Figures S1 and S2 in the revised manuscript as supporting evidence to confirm the effect of TAM on ROS.

(Figure S2; lines 99-100).

Minor:

Comment:

line 210: CHK2 inhibition would reduce BQ expression or activity? The sentence should be checked.

Response:

Thank you for your comment. We have revised the sentence.

(lines 218-219)

Round 2

Reviewer 1 Report

The authors have provided satisfactory revised material and explanation. This reviewer is very impressed with the author's efforts.